# Advances in Biomaterial-Mediated Gene Therapy for Articular Cartilage Repair

**DOI:** 10.3390/bioengineering9100502

**Published:** 2022-09-24

**Authors:** Wei Zhu, Tong Niu, Zhanqi Wei, Bo Yang, Xisheng Weng

**Affiliations:** 1Department of Orthopaedics, Peking Union Medical College Hospital, Chinese Academy of Medical Sciences & Peking Union Medical College, Beijing 100730, China; 2Department of State Key Laboratory of Complex Severe and Rare Diseases, Peking Union Medical College Hospital, Chinese Academy of Medical Sciences & Peking Union Medical College, Beijing 100730, China

**Keywords:** biomaterial, gene therapy, cartilage repair, virus vector

## Abstract

Articular cartilage defects caused by various reasons are relatively common in clinical practice, but the lack of efficient therapeutic methods remains a substantial challenge due to limitations in the chondrocytes’ repair abilities. In the search for scientific cartilage repair methods, gene therapy appears to be more effective and promising, especially with acellular biomaterial-assisted procedures. Biomaterial-mediated gene therapy has mainly been divided into non-viral vector and viral vector strategies, where the controlled delivery of gene vectors is contained using biocompatible materials. This review will introduce the common clinical methods of cartilage repair used, the strategies of gene therapy for cartilage injuries, and the latest progress.

## 1. Introduction

Articular cartilage is indispensable in joint movement; its injury and the subsequent osteoarthritis that is secondary to trauma or degeneration can be highly detrimental to patients’ quality of life [1]. Chondral lesions are not self-repairing due to limits in the capabilities of cartilage tissue, making its treatment a contemporary challenge in orthopedics. In recent years, we have witnessed a rise in cartilage injuries secondary to competitive sports and traffic accidents. The increasing awareness of health and participation in sports is, however, coupled with more sports injuries, causing wide concern in sports medicine.

Cartilage injuries are currently treated medically and surgically. During the initial stage, medicine and rehabilitation may be used among patients with mild injuries. Microfracture and autologous chondrocyte implantation (ACI) may also be used in selected patients [2]. However, joint replacement is usually indicated for patients in end stages of the disease with severe cartilage damage and osteoarthritis [3]. Despite the definitive efficacy that has been proven for arthroplasty, this surgical intervention is not considered ideal due to its high medical costs, the risks of postoperative periprosthetic joint infections (PJIs), deep venous thromboses, prostheses loosening, and the limited lifespans of implants [4,5].

Developments in tissue and genetic engineering have created new perspectives for the treatment of cartilage injuries in the past decade. Compared to traditional surgical intervention, tissue and genetic engineering focuses on complete regeneration of articular cartilage [6,7], with the potential to regenerate tissue, which is comparable to human cartilage in its quality and function. Considering the nature of cartilage’s lack of ability to self-repair or regenerate, the application of tissue and genetic engineering in cartilage repair becomes more appealing. Three core factors form the basis of tissue engineering for cartilage repair: seed cells, scaffold material, and growth factors [8]. We will not further discuss seed cell selection, as there are existing literature reviews on this topic [9,10]. Scaffold material plays an important role in tissue engineering, as it provides the environment and platform for seed cells as an extracellular matrix, and supports the growth of new tissues. Ideal scaffold material should demonstrate no immunogenicity, good biocompatibility, satisfactory biomechanical features, and be easy to produce. In recent years, the development of scaffold material has become popular in the field of cartilage repair [11]. Moreover, cellular factors are also indispensable in chondral or osteochondral repair. While promoting cartilage repair, specific cellular factors may skew stem cells in the direction of chondrocyte differentiation [12]. Previous mainstream studies incorporated growth factors directly into scaffold materials, but reported limited efficiency as a result of rapid degradation, unstable repairing outcomes, etc. [12]. The advances in genetic engineering now allow entry of exogenous genes into the cells to express various growth factors that promote cartilage repair (Figure 1). The combination of tissue engineering and genetic engineering has undoubtedly propelled the development of cartilage repair techniques. In this literature review, we will discuss the progress as well as the pros and cons of tissue and genetic engineering in the treatment of cartilage injury.

## 2. Cartilage Repair and Approaches

### 2.1. Microfracture

Microfracture has been the most common method of cartilage repair for a long time [14]. The mechanism behind microfracture is stimulating marrow by drilling through subchondral tissues that connect bone marrow, thereby promoting cellular components such as MSCs to access and repair the damaged cartilage. Previous studies, however, found that the fibrocartilage produced after microfracture was mainly type I collagen, which is different from native hyaline cartilage (type II collagen) in terms of its biochemical and biomechanical characteristics [15]. Cartilage of different types may vary tremendously in their functions. Cartilaginous tissues can be divided into three categories, based on composition and function: fibro-, elastic and hyaline cartilage. Fibrocartilage is mainly found in tendinous ligaments, fibrous rings, and in menisci, due to its high tensile resistance. Elastic cartilage contains a large amount of elastic fibers, consisting of auricular and epiglottal cartilage. Hyaline cartilage is the main component of joint cartilage, and it is also found in tracheae, bronchi, and ribs. Compared with other types of cartilage, hyaline cartilage mainly plays roles in assisting joint motions, absorbing stress, and increasing the buffer. Imaging-based and patient-reported outcome-based studies have both revealed high failure rates for microfracture operations, which may be related to the function of fibrocartilage that is generated after repairing [16,17]. Although microfracture remains the gold standard in cartilage repair, explorations to enhance its efficacy and create new therapeutic methods never cease.

### 2.2. Osteochondral Implantation

Both osteochondral autograft transfer (OAT) and osteochondral allograft transfer (OCA) are choices for performing osteochondral implantation. OAT aims to transfer autologous osteochondral tissue from a non-weight-bearing area to a weight-bearing area, promoting integration without inducing rejection. Compared to microfracture, OAT directly transfers hyaline cartilage to the injured area, with much better confirmed long-term efficacy. In an RCT, Solheim et al. [16] enrolled 40 patients who were randomly assigned to receive OAT or microfracture surgery. The results showed that the mean Lysholm score was significantly higher in the OAT group than in the microfracture group at 12 months, median 5 years, median 10 years, and minimum 15 years. However, autografting as a source remains a limit for OAT. Previous studies reported a 1.6–12.8% rate of donor site morbidity [17,18], which was further elevated if the donor site was >3 cm^2^ in area [19]. Therefore, the OAT technique is mainly used to treat small cartilage injuries, specifically with defect areas < 2 cm^2^ [20].

OCA is suitable for treating larger cartilage injuries compared to OAT, as it does not require autologous tissue while yielding similar efficacy. The survival rate after OCA was shown to be 82% at 10 years, and 66% at 20 years after transplantation, in a study by Levy et al. [21]. Another study with a mean follow-up time of 5.5 years used a 5-item categorical evaluation scale (extremely satisfied, satisfied, somewhat satisfied, somewhat dissatisfied, dissatisfied) to assess the efficacy of OCA. The results showed an overall satisfaction rate of 88.1% after OCA, and this rate remained constant over the follow-up period [22]. However, extensive application of the OCA technique is limited by high treatment costs, difficulties in obtaining allografts, and potential risks of communicable diseases related to the surgery [23,24].

### 2.3. Autologous Chondrocyte Implantation (ACI)

As a cell-based cartilage repair treatment, ACI is a two-stage surgery involving harvesting and implantation: chondrocytes harvested from a low-weight-bearing joint area are expanded in vitro, then re-implanted back to the defect area in a second operation. ACI has been shown in previous studies to provide enduring results, even in large defects (>4 cm^2^), with satisfying 10-year outcomes in terms of clinical examination and functional performance. Minas et al. [25] performed long-term follow-ups on 210 patients after ACI surgery with an average cartilage defect area of 8.4 ± 5.5 cm^2^; the average follow-up time was 12 years. The results of this study showed good efficacy of ACI. Another study by Ogura et al. [26] reported 20-year follow-up outcomes of patients undergoing ACI procedures, with an average defect area of 11.8 cm^2^. The overall survival rate was 63% after ACI surgery. Patient-reported outcomes also indicated that clinical scores improved significantly, and that the effect was sustained after 20 years postoperatively. The most common adverse event reported after ACI was hypertrophy of the periosteal flap used to seal the implanted chondrocytes over the lesion [26]. Additionally, surgical complexity, cell leakage, and high debridement rate after the initial operation all negatively affected the outcomes of ACI [20,27,28].

Matrix-induced autologous chondrocyte implantation (MACI) was introduced as the latest ACI generation, with improvements in the above-mentioned drawbacks. MACI, as a scaffold-plus-cell-based repair technique, is also performed in a two-stage setting, where chondrocytes are cultured in a type I/III collagen scaffold matrix after harvesting and implanted using fibrin glue [29,30]. The efficacy of MACI has already been proven in multiple clinical trials [31,32,33]. Evidence showed minimal risk of developing graft hypertrophy in the MACI-treated patients, but clinical efficacy has been comparable in the MACI- and ACI-treated groups. The efficacy was not different between the two treatment arms, both at the 1-year and 10-year follow-ups [34,35,36]. Further studies are still necessary to substantiate the advantage of MACI. Since both ACI and MACI are two-stage procedures, it is undeniable that a one-staged surgery would be more appealing to both surgeons and patients.

### 2.4. Autologous Matrix-Induced Chondrogenesis (AMIC)

Despite their similarities, AMIC acts with a different mechanism from MACI. AMIC is a single-stage procedure that combines biomaterial with microfracture surgery, where solid acellular type I/III collagen membrane is applied to the cartilage defect area after microfracture. Although both MACI and AMIC entail implantation of a biomaterial matrix, AMIC does not include the use of autologous articular chondrocytes, and MACI does not involve the penetration of subchondral bone (microfracture). An RCT compared the efficacy between AMIC and microfracture-based cartilage repair on 38 patients with an average cartilage defect area of 3.4 cm^2^ in the knee joint. Patients were randomized to receive sutured AMIC, glued AMIC, or microfracture. No differences were found among the three surgical groups regarding their 1-year and 2-year modified Cincinnati scores and their International Cartilage Repair Society (ICRS) scores [37]. The 5-year Cincinnati score remained stable in both AMIC groups and significantly decreased in the microfracture group, as revealed by a later study by the same research group; however, no significant difference was found in the ICRS score among the three treatment groups [38].

Since AMIC is still based on microfracture surgery, mainly the fibrous cartilage would be repaired during this procedure. Additionally, its long-term efficacy is yet to be validated in future studies.

The different surgical therapies are compared in Table 1.

## 3. Biomaterial-Mediated Gene Therapy in Cartilage Repair

Traditional tissue engineering is largely based on the strategy of using cells, supportive biomaterials, and bioactive factors to construct the desired tissue or to directly repair the defect. The main research direction in tissue engineering is to induce differentiation in a certain direction, and to transport growth factors to the lesion site in order to stimulate repair [39]. Previous studies also made attempts to incorporate growth factors into the scaffold of the biomaterial, in order to induce chondrocyte growth [40,41]. Limitations to this approach were also raised during research, taking the example of bone morphogenetic proteins (BMPs), a potential chondro-inductive bioactive factor, whose safety is now being questioned due to heterotopic ossification, immunogenicity, osteoclastic activation, and potential carcinogenicity [39,41]. Compared to incorporating bioactive factors directly into the biomaterial, genetic therapy might act in a more physiological, effective, and safer manner.

Genetic therapy, on the other hand, aims to transfer exogenous genes into target cells in order to induce endogenous gene expression to complete treatment. It has been extensively used in cartilage repair in recent years. Genetic editing is used to promote stable expressions of various growth factors for osteochondral regeneration. Meanwhile, biomaterial mediation could further localize the effect of genetic therapy by providing a scaffold to limit the genetically modified cells and vectors to within the targeted region; this increases the precision of the treatment, and makes it an ideal approach for treating osteochondral injuries.

Genetic treatment may be categorized into virus- and non-virus-based gene transfection. Virus-based gene transfection has been extensively proven in studies to have high transfection rates and continuously stable gene expressions. Its limitations include its limited packaging capacity, potential immunogenicity and carcinogenicity, and difficulties in scaling it up [42,43]. Non-virus-based gene transfection, on the other hand, is both safe and easy to scale up [44,45], but with intermediate transfection rate and gene expression efficacy [7]. In this section, we will summarize the use of viral and non-viral gene delivery vectors in cartilage repair.

### 3.1. Non-Viral Gene Delivery System

Vectors used in the non-viral gene delivery system mainly include lipid-based vectors, peptide and protein vectors, and polymeric vectors. Vector-free delivery systems have also been studied under this category (Figure 2).

#### 3.1.1. Lipid-Based Vectors

Lipid-based vectors, or liposomes, are widely applied carriers in the nano-drug delivery system. Liposomes are sealed spherical vesicles consisting of phospholipid bilayers, and are capable of protecting the genetic materials from degradation during transfection. Therefore, they are well suited for mediated genetic therapy [46]. Their special structure confers advantages such as low toxicity, high biocompatibility, biodegradability, good target gene loading capacity, and easy preparation and modification. Based on their electronic charge, liposomes may be divided into cationic, anionic, and neutral liposomes. The cationic liposome, also known as lipofectamine, is capable of transfecting not only cationic DNA, but also RNA [47], making it the most commonly used lipid-based vector at present.

Many studies have confirmed the feasibility of transferring various growth factors (IGF-1, FGF-2, and TGF-β) via liposomes into the chondrocytes to promote repair. Li et al. [48] developed a PLGA/fibrin gel hybrid scaffold to load lipofectamine/pDNA-TGF-β1 complexes and mesenchymal stem cells (MSCs) as a cartilage-mimetic tissue platform, which demonstrated good cartilage repairing. Other studies reported different outcomes using liposome-combined vectors. Lolli et al. [49] constructed a fibrin/hyaluronan (FB/HA) hydrogel scaffold to deliver antimiR-221 to the injured area, and compared antimir-221 delivery rates and repair outcomes with and without lipofectamine. The results showed a significant two-fold increase in the amount of repaired cartilage, which contained abundant type II collagen, using FB/HA loaded with antimiR-221/lipofectamine, compared to that without lipofectamine.

#### 3.1.2. Polymeric Vectors

Polymeric vectors and liposomes are the top two common non-virus-based gene delivery vectors, and have long been the gold standard for transfection of this kind. When compared to liposomes, polymeric vectors embody satisfactory variability and stability through the regulation of synthesis processes [50]. Polyethylenimine (PEI), poly (lactide-co-glycolide) (PLGA), and chitosan are all common polymeric vectors, among which PEI is the one most commonly used. Previous studies have proven its effectiveness in inducing stem cell differentiations into different cell lines after transfection [51,52], including chondrogenesis [53,54]. Additionally, researchers have made attempts to use PEI to transfer SOX9 and anti-Cbfa-1 siRNA to MSCs simultaneously, in order to enhance chondrogenesis [55]. However, studies also warned about the cytotoxicity of PEI on stem cell differentiation [56].

PLGA, a common scaffold material, can also perform the role of vectors in gene transfection. Shi et al. [57] devised a poly (L-lactic-co-glycolic acid) (PLLGA) scaffold, and utilized the PLGA vector to incorporate bone morphogenetic protein 4 (BMP-4) into rabbit adipose-derived stem cells (ADSCs) by transfection for cartilage repair. Results of this study showed good repair efficacy and outcomes using the PLGA-based scaffold and transfection vector in full-thickness articular cartilage defects, featuring a large amount of regenerated hyaline cartilage. Another widely used polymeric vector is chitosan; it exhibits good biocompatibility, low cytotoxicity, biodegradability, and no immunogenicity. Wang et al. [58] designed a composite construct comprising bone marrow mesenchymal stem cells (BMSCs), fibrin gel, and PLGA sponge. Chitosan chloride was employed as the vector to transfect transforming growth factor-β1 (TGF-β1) into BMSCs. In vivo experiments resulted in successful repair of leporine cartilage defects by the composite constructs. Histological examination confirmed a similar amount and distribution of type II collagen and glycosaminoglycans in the regenerated cartilage as those in hyaline cartilage. Effectiveness in cartilage repair has also been reported using nanohydroxyapatite (nHA) [59], composite polymeric vectors, etc. [60].

#### 3.1.3. Peptide and Protein Vectors

Another approach used to perform targeted gene delivery is through DNA-carrying peptides. Peptides are short amino acid chains of different structures with various physiological functions; these can be utilized as a part of a gene delivery system to optimize transfection. Peptides are incorporated to overcome certain systemic barriers; for example, cationic peptides with basic residues such as lysine or arginine could enhance affinity when binding to nucleic acids to form nanoparticulate complexes. Peptide and protein vectors are utilized based on their high stabilities and binding capacities, as well as their biodegradability and low toxicity. However, similar to other non-viral vectors, the transfection rate of peptide and protein vectors needs to be further improved in order to be comparable to that of viral vectors.

In a rabbit cartilage model study by Li et al. [61], a PLGA scaffold was constructed mainly with fibrin gel and mesenchymal stem cells, and a poly-l-lysine (PLL) vector was used to transfect the TGF-β1 gene into MSCs for cartilage repair. Results of this study showed that neo-cartilage could be regenerated at the lesion site, with abundant subchondral deposition of type II collagen and glycosaminoglycans. Research also discovered specific peptide sequences (for example, chondrocyte-affinity peptide (CAP)) that could target certain membrane receptors in chondrocytes. Attempts have also been made to combine the peptide vector with other carriers; for example, a CAP-PEI complex carrier designed by Pi et al. [62] demonstrated better transfection rate compared to PEI alone during in vivo experiments. It has the potential to become a cartilage-specific vector for cartilage disorders. Despite the current performance of peptide and protein vectors in studies, and their low toxicity profiles [39,63], their use in cartilage repair remains relatively underexplored, warranting further validation studies in this area.

#### 3.1.4. Vector-Free Delivery Systems

Vector-free delivery systems denote the use of multiple techniques, such as electroporation, microinjection, sonoporation, and hydrodynamic gene transfer. Due to low transfection rates and a lack of tissue-specificity, microinjection and hydrodynamic gene transfer are rarely used for gene therapy in cartilage repair [64].

Electroporation is a commonly used vector-free technology that temporarily enhances the permeability of cell membranes using pulses of high-voltage electricity, in order to promote the uptake of exogenous molecules such as DNA, RNA, or nucleic acids [65]. Nucleofection by electroporation has been performed successfully on primary chondrocytes in a high-throughput format [66]. In a study by Im et al. [67], SOX trio was transfected via electroporation into adipose stem cells (ASCs), and greatly enhanced chondrogenesis. In a study by Khoury et al. [68], electroporation was used to transfect interleukin-10 in a collagen-induced arthritis murine model. The study failed to show sufficient therapeutic efficacy despite a relatively high transfection rate, due to very unstable genetic expression that was observed during the study.

Sonoporation is another commonly used vector-free technology. It refers to the formation of small pores in cell membranes using ultrasound for the transfer of nucleic acid materials. An in vivo study revealed highly efficient BMP-6 transfection into MSCs via sonoporation to improve fracture healing [69]. This technique is also being applied to transfect genes into the intravertebral disks in some studies [70]. The effect of using sonoporation in cartilage injury repair has not been reported in the literature, and future studies are warranted to further demonstrate its efficacy.

Despite having lower costs and better safety profiles compared to virus-based therapy, solutions to low gene expressions and transfection rates are yet to be found in order to revolutionize non-virus-based genetic therapy [43]. Additionally, the efficacy of combining vector-free delivery systems and biomaterials to improve cartilage repair outcomes has not been extensively discussed in the literature. The application of this combined technique also warrants future exploration in this field.

Different non-viral gene delivery systems are compared in Table 2.

### 3.2. Virus Gene Delivery Vectors

Due to its high efficiency in cell infection and its ability to integrate with the host cell genome, viral vectors are a commonly applied delivery system for gene therapy, such as retroviruses/lentiviruses, adenoviruses, adeno-associated viruses (AAVs), and baculoviruses.

#### 3.2.1. Retrovirus/Lentiviral

Retroviruses can integrate their own genes into the host chromosome, thus ensuring the continuity of the integrated genes that can be expressed in the cell [71]. They have the advantages of a wide spectrum of infection, effective infection of cells at the dividing and resting stages, and long-term stable expressions of exogenous genes [72]. Therefore, retroviruses are a powerful tool for introducing exogenous genes. Clinical trials have been conducted to achieve gene therapy by transfecting the synovial cells of inflammatory joints with retroviruses that express IL-1 receptor antagonist (IL-1 Ra), and then inserting these cells into the joint cavity of rheumatoid arthritis [73]. However, retroviruses will preferentially integrate the genes that are carried into the transcription starting point as well as highly expressed genes, which will lead to tumor side effects [74], such as leukemia in x-linked patients with severe combined immune deficiency [75]. Lentiviral vectors are mostly integrated into sites that are far from the transcription starting point. Thus, compared with retrovirus vectors, lentiviral vectors appear to be less likely to cause cancer, and may be safer for clinical use.

Many studies focused on generating gene-modified scaffold-mimicking cartilaginous extracellular matrix (ECM) through retrovirus/lentivirus-based methods, in order to improve cartilage repair [76]. A gene-modified silk cable-reinforced chondroitin sulfate–hyaluronate acid–silk fibroin (CHS) hybrid scaffold was developed to reconstruct the fibrocartilage layer [77]. Mesenchymal stem cells (MSCs) were able to distribute uniformly throughout the scaffold with the lentiviral-mediated transforming growth factor-*β*3 (TGF-*β*3) gene, and showed chondral differentiation. Polycaprolactone (PCL)-hydroxyapatite (HA) scaffold [78] and poly(e-caprolactone) scaffold [79] were also explored to enable lentiviral-mediated TGF-*β*3 gene overexpression, and showed promising cartilage defect repair. Moreover, Lee et al. [80] used retroviruses to transfect the SOX gene into adipose stem cells and compound the transfected cells with fibrin hydrogel. In the rat model of cartilage injury, it was found that the composite material could promote the repair of articular cartilage defects and delay the degeneration of arthritis. Inducing overexpression of the IL-1 receptor antagonist (IL-1Ra) in MSCs via scaffold-mediated lentiviral gene delivery was also able to enhance the long-term success of therapies for cartilage injuries or osteoarthritis by resisting the IL-1-induced upregulation of matrix metalloproteinases [8].

#### 3.2.2. Adenovirus

Adenovirus (AdV) is a double-stranded DNA without an envelope, containing approximately 26–48 kBP in its genome [81]. More than 60 human adenoviruses have been identified, with adenovirus serotypes 5 (Ad5) widely used as a gene delivery vector [81]. Adenoviruses have low or no toxicity in humans, and high transduction efficiency in both mitotic and non-mitotic cells [81,82]. Moreover, adenoviruses have a very low risk of insertion mutation because they cannot be integrated into the host genome; however, they also have the disadvantage of not being able to express the carrying genes for long [81,82].

Recombinant adenoviruses can be successfully transfected into cells derived from bone marrow fluid, such as BMSCs, ASCs, and induced pluripotent stem cells [83]. Adenovirus-mediated Sox9 gene transfer of bone marrow mesenchymal stem cells was able to induce chondrogenesis in a PGA scaffold [84]. In a rabbit model with full-thickness cartilage defects, the PGA scaffold and BMSCs with Sox9 transduction-grafted joints showed more newly formed cartilage tissue and hyaline cartilage-specific extracellular matrix, and greater expressions of several chondrogenesis marker genes. Another study synthesized a chitosan/silk fibroin (CS/SF) porous scaffold with bone-marrow-derived mesenchymal stem cells (BMSCs), using transfection with recombinant adenovirus containing C-type natriuretic peptide (CNP) gene; this also showed good chondrogenic differentiation ability and promising cartilage lesion repair in a rat model [85].

The biggest obstacle to the clinical application of adenoviruses is the strong humoral and cellular immune responses caused by them [86]. Researchers developed “gutless” vectors, containing only virus terminal repeat sequences and packaging sequences, in order to minimize the immune response [87]. However, the production of “gutless” adenovirus vectors is more complicated due to the absence of most viral components, and the need for auxiliary plasmids or viruses [88].

#### 3.2.3. Adeno-Associated Virus

Adeno-associated virus (AAV), a prospect for widely applicable gene vector delivery, is a low-pathogenic parvovirus. Its replication requires helper viruses, for example, adenoviruses. Its genome is linear, single-stranded DNA with a size of about 4.7 kB. AAVs may provide long-term transgene expression in numerous dividing and non-dividing cells without triggering potent host immune responses, which make them non-pathogenic to humans. These advantages enable AAVs to demonstrate clinical potential in the treatment of tumors, hemophilia, lipoprotein lipase deficiency and other diseases. Researchers have developed an AAV vector that expresses TNF antagonists for the treatment of RA, which has successfully entered phase I and II clinical trials.

Some studies have used intra-articular injections of recombinant adeno-associated viruses to affect cartilage metabolism through cytokines such as IL-1β, L-1R, and TNF-α, in order to achieve the effect of cartilage injury repair. However, the transfection efficiency of the simple virus is not high. Some scholars have combined recombinant adeno-associated virus with cells and biological materials, in order to improve transfection efficiency and repair effects. Jagadeesh et al. [89] found that recombinant adeno-associated virus and biocompatible mechanostable poly(E-caprolactone) (PCL) films grafted with poly(sodium sulfonate) (pNaSS) could still achieve 90% transfection efficiency after 21 days, with no biotoxicity detected. Fibrin scaffolds can also serve as long-term releasers of recombinant adeno-associated virus vectors [90]. These results suggest that scaffold-guided gene transfer offers strong systems to develop promising therapeutic options for the treatment of articular cartilage defects. Moreover, Ana et al. [91] developed poly (ethylene oxide) (PEO) and poly(propyleneoxide) (PPO) polymeric (PEO−PPO−PEO) micelles to control release-transferring SOX9 rAAV gene vectors. Controlled delivery of recombinant rAAV via polymeric micelles overexpresses the levels of SOX9, leading to increased proteoglycan deposition and a stimulated proliferation of OA chondrocytes. In 1-year minipig cartilage defect models [92], alginate hydrogel guided with rAAV-mediated IGF-1 overexpression was able to enhance long-term cartilage repair and protection against perifocal osteoarthritis without deleterious or immune reactions. These results suggest that hydrogels, micelles, and scaffolds have great potential in gene therapy mediated by recombinant adeno-associated viruses.

#### 3.2.4. Baculovirus

Baculoviruses can naturally infect insect cells, and have been widely employed to transmute many mammalian cells. They have demonstrated efficient gene delivery-mediated expression of growth factors (TGF-b1, IGF-1, and BMP-2) to therapeutic levels in chondrocytes, thus showing their potential for application in cartilage tissue engineering [93]. Chen et al. [94] developed baculovirus-transduced chondrocytes, and then seeded them in PLGA porous scaffold, which consequently demonstrated chondrogenic abilities. However, due to the non-replicating nature of baculoviruses, they mediate transient (<7 days) transgenic expressions; this hinders the application of baculoviruses in situations where ongoing expression is required [95,96].

Different virus gene delivery systems are compared in Table 3.

## 4. Limitations and Perspectives

Gene therapy is used to promote tissue regeneration by transfecting specific genes into target cells and upregulating their expressions, in order to increase the synthesis of extracellular matrix and accelerate the differentiation of target cells to target cells [97]. It has made significant breakthroughs in cartilage repair. The expressions of genes with specific functions in cartilage defects can promote the regeneration of cartilage [86]. One study found that the direct implantation of stem cells into the joint after genetic modification in vitro can lead to the successful repair of cartilage defects, by upregulating cartilage differentiation-promoting genes and downregulating inhibitory genes [83].

The problem that often needs to be solved in the process of using gene therapy for articular cartilage defects is finding means to avoid the flushing and diluting of injected transfected cells from the synovial fluid. Researchers have used different cell scaffolds as modified cell carriers, and co-injected them into the joint cavity [85,87]. When the scaffolds were degraded, the cell-coated carriers were adsorbed onto the defect site, thereby solving this problem. A combination of gene therapy and scaffold material can greatly improve the efficiency of gene transfection, and promote cartilage regeneration and repair. Recent studies [85,87] have shown that a combination of appropriate delivery vehicles, genes, target cells, and scaffolds can effectively promote hyaline cartilage growth. However, although gene therapy for alleviating bone and joint-related diseases has been widely verified theoretically, there are remaining problems that must be overcome, such as the apoptosis of target cells during gene therapy, the spread of diseases, etc.

The choices of genetic targets are also worthy of discussion for cartilage repair. While previous studies largely focused on classic targets such as BMP, SOX, or TGF-β, attention is being paid to the relationship between miRNA and cartilage repair [98,99]. There is potential to improve repair efficacy by combining the application of biomaterials and gene therapy with multiple targets.

Exploring and developing novel biomaterials with good biocompatibility and enhanced interactions between the materials and transgenic cells are crucial to advancing gene therapy. In addition, the integration of a composite graft with the host tissue is important to investigate further. Fortunately, the combined use of mesenchymal stem cells, cell-stimulating factors, and cyto-scaffolds may offer new avenues to address these challenges. Finally, the most serious challenge is to translate the results from successful animal experimental research into the clinic. Today, there are a wide variety of cartilage tissue engineered products, but only a few are available for clinical trials. The possible reasons for this include product quality control, stability, cost, safety, and patent issues. Mature clinical applications of extraction and transfection still need further research.

The applications of gene therapy and tissue engineering have accelerated the therapeutic development of cartilage repair. Despite treatment, patients’ outcomes are still affected by many other factors. Current treatment options mostly focus on cellular biological traits as modifiable targets. The mechanical characteristics of the cartilage also warrant targeted approaches, as they are crucial to articular functions. These include the reconstruction of their biomechanical characteristics and their efficiently lubricated surfaces [100,101]. Articular cartilage is composed of water, type II collagen, and proteoglycan. The composition and structure of collagen differ from the superficial zone, transitional zone, to the radial zone, and these differences influence cartilage function [100]. There is a paucity of evidence on the integral function of regenerated cartilage via current gene therapy and tissue engineering, such as its lubricative performance and mechanical features, and this warrants further study.

## 5. Conclusions

The repair of cartilage injuries will remain under the research spotlight in the near future. The combination of gene therapy and tissue engineering will undoubtedly play an important role in this field, with promising outlooks for both non-virus-vector- and virus-vector-based techniques. As ongoing studies are limited to in vitro studies or on animal models in nature, clinical translations of these approaches await further exploration. Moreover, current repair techniques still have some drawbacks; while reconstructing cellular biological characteristics is necessary to restore function, other factors may affect the therapeutic outcomes [102]. Perhaps it would be wiser to think “out of the box” in order to facilitate advances in achieving true regeneration of articular cartilage, for example, in modifying biomechanical properties, or in applying alternative gene editing methods such as introducing non-coding RNA into current gene therapy practices.

## Figures and Tables

**Figure 1 bioengineering-09-00502-f001:**
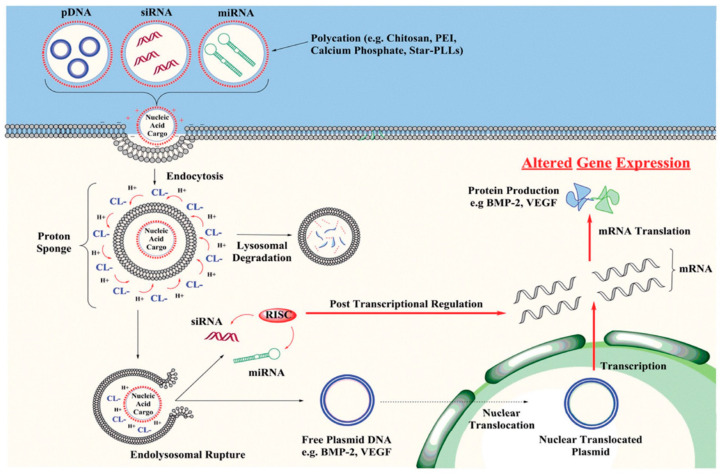
Non-viral vectors delivering pDNA, siRNA or miRNA into target cells by endocytosis or membrane fusion. Once inside the cells, most vector–nucleic acid complexes become trapped in an endosome. The nuclear acid cargo is then released to enter the nucleus for integration and expression [13]. Copyright: Wiley Online Library.

**Figure 2 bioengineering-09-00502-f002:**
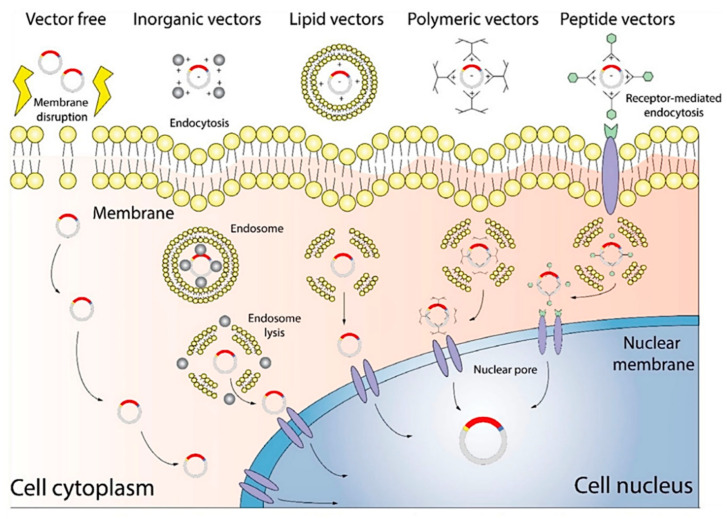
Diagram of a non-viral gene delivery system. Vector-free technology delivers target genes by altering the cellular membrane, while other vectors deliver target genes through interactions with the membrane, for example, endocytosis [39]. Copyright: Wiley Online Library.

**Table 1 bioengineering-09-00502-t001:** Comparison of different surgical therapy methods for cartilage injuries.

Therapy Method	Indications	Cartilage Source	Advantages	Disadvantages
Microfracture	Small cartilage injury-defect area < 2 cm^2^	N/A	Low cost; technically easy	Repaired by fibrous cartilage; questionable long-term efficacy
Osteochondral implantation				
Osteochondral autograft transfer (OAT)	Small to medium cartilage injury-defect area 2–4 cm^2^	Autograft	Repaired by hyaline cartilage; fast graft integration	Donor site morbidity; potential risk of disease transmission
Osteochondral allograft transfer (OCA)	Medium to large cartilage injury-defect area > 2 cm^2^	Allograft	Repaired by hyaline cartilage; can treat large cartilage injuries;	Allograft availability; high cost
Autologous chondrocyte implantation (ACI)	Medium to large cartilage injury-defect area > 2 cm^2^	Ex vivo cultured autologous chondrocytes	Can treat large cartilage injuries	High cost; two-stage operation; Graft hypertrophy
Matrix-induced autologous chondrocyte implantation (MACI)	Medium to large cartilage injury-defect area > 2 cm^2^	Ex vivo cultured autologous chondrocytes	Can treat large cartilage injuries	High cost; two-stage operation
Autologous matrix-induced chondrogenesis (AMIC)	Small cartilage injury-defect area < 2 cm^2^	N/A	Superior repair tissue quality compared with microfracture; technically easy	Repaired by fibrous cartilage; questionable long-term efficacy

**Table 2 bioengineering-09-00502-t002:** A comparison of different non-viral gene delivery systems used in biomaterial-mediated gene therapy.

Types	Subtypes	Biomaterials	Genes	Technology Readiness Levels	Advantages	Disadvantages
Lipid-based vectors	Lipofectamine [48,49]	PLGA/fibrin gel hybrids scaffold; fibrin/hyaluronan hydrogel scaffold	TGF-β1; antimiR-221	In vitro; in vitro	High biocompatibility; Biodegradability; Good capacity; Ease of large scale production	Cytotoxicity; Low stability; Low half-life
Polymeric vectors	PEI [55]; PLGA [57]; Chitosan [58]; nHA [59]	PLLGA scaffold; fibrin gel and PLGA sponge; alginate hydrogels	SOX9 and anti-Cbfa-1 siRNA; BMP-4; TGF-β1; TGF-β3 and BMP2	In vitro; in vivo; in vivo; in vitro	Satisfying variability; High stability; Easy to incorporate into biomaterials	Cytotoxicity; Immunogenicity
Peptide and protein vectors	PLL [61]	PLGA scaffold	TGF-β1	In vivo	High stability; High binding capacity; Biodegradability; Low toxicity	Low transfection efficiency

Abbreviations: PEI, polyethylenimine; PLGA, poly (lactide-co-glycolide); PLLGA, poly (L-lactic-co-glycolic acid); nHA, nanohydroxyapatite; PLL, poly-l-lysine.

**Table 3 bioengineering-09-00502-t003:** A comparison of different virus vectors used in biomaterial-mediated gene therapy.

Vectors	Genome	Integratable or Not	Maintaining Expression	Immune Response	Biomaterials	Genes	Technology Readiness Levels
Retrovirus/Lentiviral	ssRNA	Random integration and stable inheritance	stable and long expression	Medium immunogenicity	CHS [77] PCL-HA [78] poly(e-caprolactone) [79] fibrin [80]	TGF-ß3 TGF-ß3 TGF-ß3 SOX	In vitro In vivo In vitro In vivo
Adenovirus	dsDNA	Unintegratable	3 weeks	High immunogenicity	PGA scaffold [84] CS/SF scaffold [85]	SOX-9 CNP	In vivo In vivo
Adeno-associated virus	ssDNA	Unintegratable	At least 6 months	Low immunogenicity	poly(E-caprolactone) (PCL) films grafted with poly(Sodium Sulfonate) (pNaSS) [89] PEO−PPO−PEO micelles [91]	Cy3 SOX9	In vitro In vivo
Baculovirus	dsDNA	Unintegratable	1 week	Low immunogenicity	PLGA porous scaffold [94]	EGFP	In vitro

## Data Availability

Not applicable.

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
