# Peer review of "Advances in Biomaterial-Mediated Gene Therapy for Articular Cartilage Repair"

_bioengineering, 2022, doi:10.3390/bioengineering9100502_

Round 1

Reviewer 1 Report

Current manuscript reviews available approaches to repair articular cartilage, before summarize advances in gene therapy in this field. It is somewhat mediocre, but with some work meaningful.

1)      The manuscript with two cited, non-original figures and one short table means that it lacks of summary of advances, possibly.

2)      The section “cartilage repair and approaches” is redundant, which has been intensively reviewed heavily.

3)      The section “biomaterials-mediate gene therapy in cartilage repair” adopts broadly-used classification: non-virus Vs virus gene delivery systems, and summarize advances in “cartilage field”. However, key questions should be answered before this summary: what are specific challenges and properties of gene therapy or gene delivery in cartilage field, except for common challenges of gene therapy or gene delivery? When talking about articular cartilage, what are its unique properties, different from other types of cartilage? Taking these challenges could upgrade this review.

4)      When tissue engineering, as well as its three core factors failed to repair articular cartilage with full functionality, it is wise and necessary to think “out of the box”: do we have to adopt different strategies? The answer is Yes, as discussed broadly[1].

Reference:

1.    Malda, J., J. Groll, and P.R. van Weeren, Rethinking articular cartilage regeneration based on a 250-year-old statement. Nat Rev Rheumatol, 2019. 15(10): p. 571-572.

Author Response

We are very grateful to you for your time in reviewing our manuscript and providing valuable feedback. Here in the attachment we have addressed the comments and suggestions kindly raised by the reviewers.

Reviewer 2 Report

This is a well-written review about biomaterial-mediated gene therapy for articular cartilage repair. I recommend it for publication after the following points are addressed.

1. The importance of the surface layer of cartilage was ignored in this review. The authors should add discussions about the lubrication properties of healthy cartilage. Several reviews (Advanced Materials 33 (18), 2005513, 2021; Polymers 13 (12), 2000, 2021) related to this point should be included.

2. The resolution of figure 2 should be improved.

3. It is unclear how PLGA (hydrophobic polymer) is involved in gene therapy. The authors should not call PLGA as the vector for gene therapy.

4. A section of conclusion is encourage to be added.

4.  

Author Response

(The authors gave the same response as above.)

Reviewer 3 Report

The review quite fully presents all the current developments in the field of gene therapy for cartilage damage. However, for the best presentation of the available approaches and their success in using and implementing them in the clinic, I can recommend the authors to compile an appropriate table that shows the methods, their degree of readiness for use in the clinic and provides a corresponding reference.

Author Response

(The authors gave the same response as above.)
